# Big Earth Observation Data Integration in Remote Sensing Based on a Distributed Spatial Framework

**Yinyi Cheng** [1,2,3,4], **Kefa Zhou** [1,2,3,4,*], **Jinlin Wang** [1,2,3,4] **and Jining Yan** [5]

[1] State Key Laboratory of Desert and Oasis Ecology, Xinjiang Institute of Ecology and Geography, Chinese Academy of Sciences, Urumqi 830011, China; chengyinyi17@mails.ucas.ac.cn (Y.C.); wangjinlin@ms.xjb.ac.cn (J.W.)

[2] Xinjiang Key Laboratory of Mineral Resources and Digital Geology, Urumqi 830011, China

[3] Xinjiang Research Center for Mineral Resources, Chinese Academy of Sciences, Urumqi 830011, China

[4] University of Chinese Academy of Sciences, Beijing 100049, China

[5] School of Computer Science, China University of Geosciences, Wuhan 430074, China; yanjn@cug.edu.cn

\* Correspondence: zhoukf@ms.xjb.ac.cn

**Abstract:** The arrival of the era of big data for Earth observation (EO) indicates that traditional data management models have been unable to meet the needs of remote sensing data in big data environments. With the launch of the first remote sensing satellite, the volume of remote sensing data has also been increasing, and traditional data storage methods have been unable to ensure the efficient management of large amounts of remote sensing data. Therefore, a professional remote sensing big data integration method is sorely needed. In recent years, the emergence of some new technical methods has provided effective solutions for multi-source remote sensing data integration. This paper proposes a multi-source remote sensing data integration framework based on a distributed management model. In this framework, the multi-source remote sensing data are partitioned by the proposed spatial segmentation indexing (SSI) model through spatial grid segmentation. The designed complete information description system, based on International Organization for Standardization (ISO) 19115, can explain multi-source remote sensing data in detail. Then, the distributed storage method of data based on MongoDB is used to store multi-source remote sensing data. The distributed storage method is physically based on the sharding mechanism of the MongoDB database, and it can provide advantages for the security and performance of the preservation of remote sensing data. Finally, several experiments have been designed to test the performance of this framework in integrating multi-source remote sensing data. The results show that the storage and retrieval performance of the distributed remote sensing data integration framework proposed in this paper is superior. At the same time, the grid level of the SSI model proposed in this paper also has an important impact on the storage efficiency of remote sensing data. Therefore, the remote storage data integration framework, based on distributed storage, can provide new technical support and development prospects for big EO data.

**Keywords:** big earth observation data; remote sensing data integration; distributed storage; SSI Model; OLC; remote sensing metadata

## 1. Introduction

Changes in the atmosphere, ocean, land, vegetation, and other factors in Earth systems affect human activities all the time. As an integrated system, the Earth includes all fields involved in the various disciplines of geoscience and information technology. Earth observation (EO) systems provide a useful source of information for humans to understand Earth systems, and also are an

indispensable research method for researchers to study Earth systems in detail. EO systems can provide continuous global multi-temporal Earth data [1]. These kinds of data can be used to describe the Earth system as a whole [2]. EO systems include Earth observation satellites, airborne remote sensing systems, and EO data receiving systems, and multiple platforms for observing Earth cooperate with each other [3]. This greater system is equipped with various types of sensors, which can implement real-time observation and dynamic monitoring of the global land, atmosphere, and ocean. As of 2017, 1738 satellites were in normal operation, including 596 EO satellites. The amount of data acquired by the Ziyuan-3 (ZY-3) satellite in 2012 was more than 10 TB per day. Advances in remote sensing technology and information technology have led to the rapid growth of remote sensing data, and global remote sensing data will eventually reach the petabyte level [4–6]. Therefore, global Earth observation already has the ability to acquire high-resolution and high-precision temporal data for the atmosphere, ocean, and land, and EO systems have entered the era of big EO data. Remote sensing data are one of the most important data sources in EO systems [7]. Big EO data are obtained from Earth observation systems, such as the Global Earth Observation System of Systems (GEOSS, http://www.earthobservations.org/index.php), the European Space Agency Copernicus Open Access system (https://scihub.copernicus.eu/), Earth Observing System Data and Information System (EOSIS, https://earthdata.nasa.gov/), and the United States Geological Survey (USGS) Global Visualization Viewer system (https://glovis.usgs.gov/).

In recent years, more attention has been paid to the remote sensing data integration from different sources. Over the past 30 years, the China Remote Sensing Satellite Ground Station has received a series of domestic and foreign satellite data, including that from Landsat, Systeme Probatoire d'Observation de la Terre (SPOT) and China & Brazil Earth Resource Satellite (CBERS). As of 2013, it has archived more than 3.3 million kinds of EO satellite data. Due to differences in orbital parameters, satellite revisit periods, spatial resolutions, and sensor types of the remote sensing data, these differences have caused some difficulties in the integration of the vast amount of remote sensing data [8]. In addition, a complete remote sensing big data information descriptive system can provide users with quick and accurate retrieval services for big EO data. Therefore, exploring an efficient remote sensing data management framework from different sources can provide a data foundation for big EO data [9]. The National Snow and Ice Data Center has transformed its remote sensing data storage method from standalone storage to an online storage mode [10]. Liu proposed a distributed integration framework for heterogeneous EO data under the OpenSearch protocol [11]. At the same time, new EO data systems, represented by the Google Earth Engine (GEE) in the United States, Data Cube in Australia, and Copernicus for the European Space Agency (ESA), etc., have achieved multi-source remote sensing data integration.

Presently, there are two modes for managing and storing remote sensing image data, namely, file management and relational database management. Due to the characteristics of remote sensing data, most remote sensing image processing software uses a file management system to organize remote sensing data [12]. However, a relational database management system (RDBMS) has the ability to manage remote sensing data by integrating complex data types [13]. Recently, a number of distributed storage technologies with the ability to manage unstructured data have provided support for big EO data. Kou proposed a strategy of using a RDBMS and Hadoop Distributed File System (HDFS) to store remote sensing data and metadata [14]. Jing proposed a storage model for distributed remote sensing data based on HBase [15]. Marek provided a distributed system for storing EO data based on HTML5 and WebGL [16]. At present, the traditional methods for managing EO data have been unable to meet the requirements of data integration in big data environments. As Ma has stated, in order to improve the sharing and interoperation of EO data, innovation of the data storage framework is necessary [17]. Therefore, we should explore an efficient multi-source big EO data integration organization model. This model should (1) have a reasonable spatial organization management model, (2) build a complete spatial information descriptive system, and (3) use an efficient spatially distributed storage framework.

Not Only Structured Query Language (NoSQL) a spatial organization and distributed database technology, can provide new research methods for processing and analyzing big EO data. In this paper, a new framework is proposed to solve the data structure problems in implementing the integration of multi-source remote sensing data. In terms of spatial management, the spatial segmentation indexing (SSI) model proposed in this paper improves the organizational management and integration efficiency of remote sensing data. For data management, this paper uses MongoDB technology to perform the distributed integration of remote sensing data and metadata. Especially, the parallel processing method of remote sensing data based on MongoDB is innovatively proposed in this paper to implement the distributed storage of remote sensing data. This not only improves the storage efficiency of remote sensing data in the database, but also provides data structure support for the parallel computing of big EO data. At the same time, the remote sensing metadata are no longer integrated via extensible markup language (XML). This method can reduce the redundancy between metadata and improve the retrieval efficiency of multi-source remote sensing data.

This paper aims to build a distributed spatial framework to integrate EO data. The structure of this article is as follows: Section 2 introduces the big EO data integration distributed framework proposed in this paper, describing it in detail, including the SSI model, remote sensing information descriptive system, and distributed storage method. The experimental data and the design of the experiment here is presented in Section 3. The storage efficiency and retrieval results of multi-source remote sensing data in different environments will be discussed in Section 4. Section 5 summarizes the paper.

## 2. Method

### 2.1. Multi-Source Remote Sensing Data Integration Framework

This paper proposes an efficient spatial data management method based on a distributed database for the integration and management of shared multi-source remote sensing data sets. This framework consists of two parts: the remote sensing data are spatially partitioned according to the SSI model, and the fragmented data are automatically associated with the descriptive information system. Then, through a distributed data center, a splitter, separating remote sensing data and metadata, is inserted into the distributed data management system for sharding storage, as shown in Figure 1.

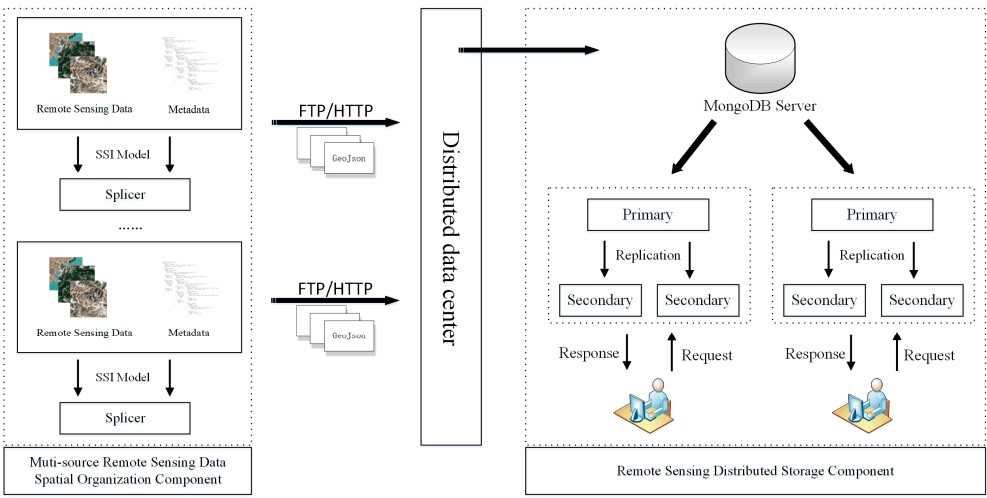

**Figure 1.** Architecture of the proposed method.

In order to improve the integration efficiency of the remote sensing data, based on the spatial indexing method of the SSI model, the remote sensing data are divided from large datasets into blocks with spatial organization. The remote sensing metadata, with the Geo-JavaScript Object Notation

(GeoJSON) format, are automatically matched with corresponding remote sensing data block. The remote sensing data block and the metadata form a splitter, which is uploaded to the distributed data center through a transmission protocol such as File Transfer Protocol (FTP) or HyperText Transfer Protocol (HTTP). In this process, the remote sensing data are transmitted in the Geotiff format and the metadata are transmitted in GeoJSON format. The distributed data center will hand over the multi-source remote sensing data records to the MongoDB server for processing. The config server uses dataset information and data fragmentation information for distribution. Each shard includes at least three members that store data, including a primary member and two secondary members. When the master database goes down, two slaves will run for election, one of which will become the master database. After the original master database recovers, it may join the current replication cluster as a slave. The horizontal expansion of the sharding mode can make more efficient use of unused computer resources, while using replication sets for sharding can reduce the time that the database cannot provide data support and achieve 100% availability. In this way, the framework can provide secure, efficient, and fast multi-source remote sensing data storage and retrieval capabilities for big EO data services.

*2.2. Spatial Segmentation Indexing Model*

The SSI model is based on the Open Location Code (OLC) spatial position latitude and longitude encoding method. The OLC was proposed by Google in 2014. The core idea is that after encoding the WGS84 latitude and longitude, the returned string can represent any area on the Earth. As shown in Table 1, as the length of the code increases, the accuracy of the regions on the Earth represented by the code increase. The height and width of the area represented by the first two codes are both 20 degrees. In these two codes, the first number represents the latitude and the second represents the accuracy, as shown in Figure 2. In the first 10-bit code, every second code added has an accuracy of 1/20 of that of the original. With the increase of the number of code bits, the target area is divided into $20 \times 20$ grids. Similarly, the first of the two-digit numbers represents the latitude and the second represents the longitude. Starting from the 11-bit code, different algorithms are used to encode and convert. This has the advantage of shortening the length of the code. For instance, one may divide the area of the 10-digit code into a $4 \times 5$ grid, where a number represents a grid. For example, we have calculated an 11-digit code, which represents the area of $1/32000° \times 1/40000°$, or the area of $3.4 \times 2.7$ square meters in the equatorial area. The basic encoding rule is completely arranged by the 26 English letters and 36 characters from 0 to 9 and uses 10,000 words from 36 languages for evaluation, and 20 characters are selected as the identifier required in the encoding. The purpose of this rule is to avoid unnecessary spelling mistakes and accidental coincidence with existing words. Since OLC is based on 20, the characters will be less than the latitude and longitude.

**Table 1.** Precision of the valid code lengths on different level.

| Level | Precision in Degrees (°) | Precision (Meter) |
|---|---|---|
| 1 | 20 | $2.2 \times 10^6$ |
| 2 | 1 | $1.1 \times 10^5$ |
| 3 | 1/20 | $5 \times 10^3$ |
| 4 | 1/400 | 278 |
| 5 | 1/8000 | 13.9 |
| 6 | $1/40000 \times 1/32000$ | $2.8 \times 3.5$ |
| 7 | $1/200000 \times 1/128000$ | $0.56 \times 0.87$ |

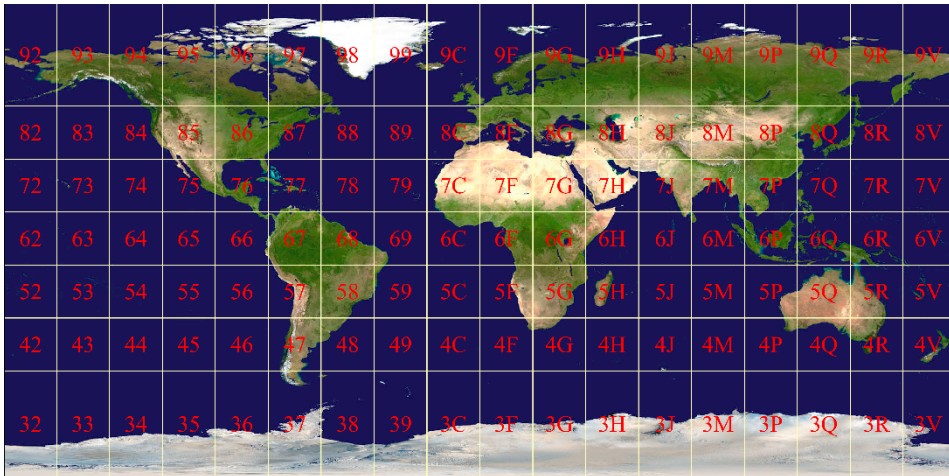

**Figure 2.** Level 0 Grid with Open Location Code (OLC) on spatial segmentation indexing (SSI) Model.

Finally, after multiple partitions, the entire Earth system can be accurately described, even down to the millimeter level. Through the conversion of offline algorithms, a string of characters can replace two or more redundant latitude and longitude coordinate points to represent a spatial index. At the same time, the conversion to latitude and longitude coordinates can provide spatial index association with the instance distance on the Earth. In addition, because models such as quadtrees are not used, there will be no errors in spatial expansion. In short, the big EO data in different spatial coordinate systems can be accurately located by a unique spatial index in order to build a system with geospatial significance.

Based on the OLC spatial segmentation indexing strategy, a spatial segmentation indexing (SSI) model for multi-source remote sensing data is proposed in this paper. The SSI model will be used as the spatial index identifier in this paper. Remote sensing data slices will no longer be identified by matching the minimum bounding rectangle (MBR) with the grid [18]. The MBR will cause the spatial index of the remote sensing data to overlap and the position to be misaligned. In this paper, the pre-processed multi-source remote sensing data are spatially matched with the SSI model, which meets the accuracy requirements. The matched code is inserted into the corresponding remote sensing data metadata. Through spatial index encoding and the conversion of remote sensing data at different spatial locations, a virtual mapping is established between the geospatial location and the spatial index. In fact, the SSI model has established spatial relationships between different types of remote sensing data and geographic entities in the Earth system. The latitude and longitude position information recording method, with higher redundancy, is hence improved, thereby increasing the speed of spatial retrieval and reducing the storage size of the remote sensing attribute data. In addition, by finding the corresponding SSI grid code through coordinates, the remote sensing data slices in the target area can be quickly located. Therefore, the SSI model provides a good spatial management and organization method for big EO data and realizes the function of fast spatial retrieval.

*2.3. Geo-JavaScript Object Notation (GeoJSON)-Based Distributed Remote Sensing Data Description Method*

The method proposed in this paper in based on metadata of remote sensing data on ISO 19115-2: 2009 and the description method is based on the GeoJSON format. It plays an important connection role in the framework, as shown Figure 3. The metadata of remote sensing data represent descriptive information about the data and usually contain multiple dimensional features, such as geographic location information, band information, the satellite access time, and the spatial resolution. However, due to the characteristics of multi-source remote sensing data, the format differences of remote sensing metadata from different sources have made it impossible to integrate multi-source remote sensing data [19]. Wang proposed to use XML as a standard to solve the exchange and sharing of metadata

from different sources [20]. The China Center for Resources Satellite Data and Applications uses the XML file format as the storage format for remote sensing metadata. For the spatial management of multi-source remote sensing data, a metadata system with real-time retrieval characteristics needs to be constructed. This paper proposes a distributed remote sensing metadata management mechanism based on the GeoJSON format. XML is a language similar to HTML. It has no predefined tags and uses document type definition (DTD) to organize data. The XML file format is huge and complex, and often requires a lot of computer resources to process the XML file. GeoJSON is used in this paper instead, which is a lightweight exchange format that supports various geographic data structures. It has the characteristics of a simple format and easy storage. It is a spatial expansion based on the JavaScript Object Notation (JSON) file format. The kernel contains many geospatial attributes, especially some important coordinate systems. Remote sensing metadata can be used to store information in a key/value structure. Meanwhile, it is a geospatial data storage format supported by MongoDB, which is used in this paper, which can provide the advantages of rapid sharding and retrieval for the distributed storage of remote sensing metadata.

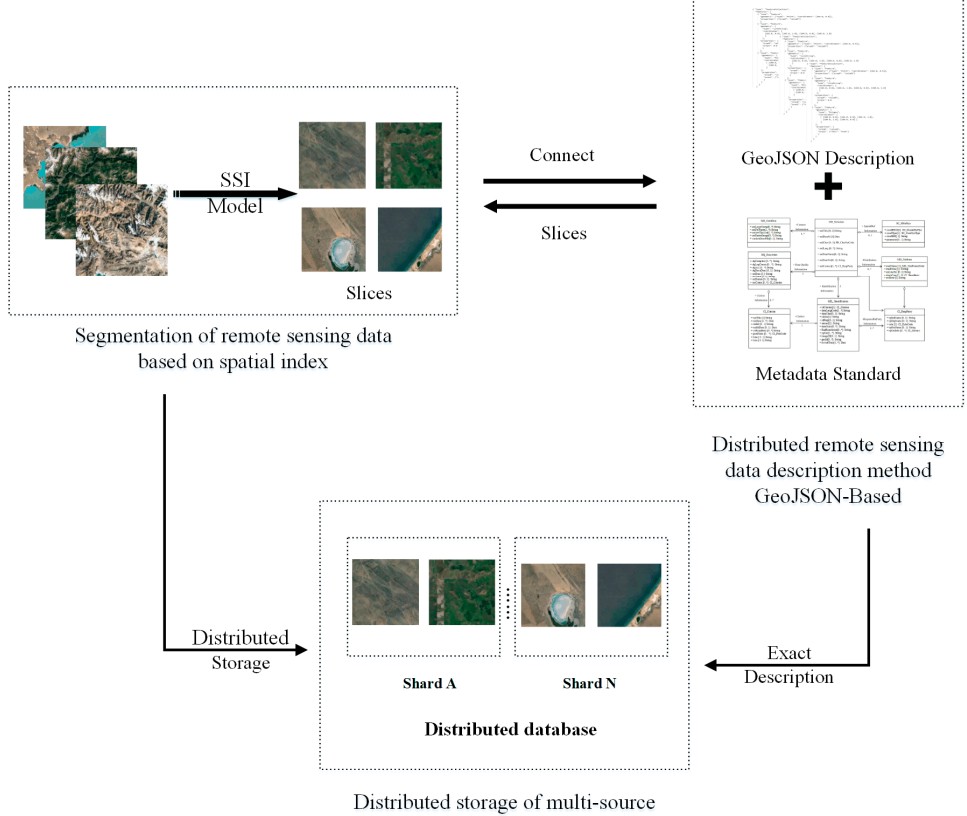

**Figure 3.** Distributed remote sensing data description method Geo-JavaScript Object Notation (GeoJSON)-Based logic diagram.

Metadata are also a kind of data. Metadata represent the most basic feature set abstracted from complex remote sensing data. Metadata are usually organized by elements, entities, and subsets. Elements are used to describe a specific feature of the dataset. Entities are a collection of metadata elements that describe similar characteristics. Subsets are collections of interrelated metadata entities and elements. This paper is based on the ISO 19115-2: 2009 geographic metadata standard. According to the characteristics of remote sensing data, an expression method combining the Unified Modeling Language (UML) and a data dictionary is used to describe the content and structure of remote sensing core metadata [21]. In practical applications, remote sensing metadata

should describe the characteristic information of the multi-source remote sensing data. Based on the above ideas, this paper mainly designs a framework model composed of 7 metadata subsets, as shown in Figure 4, which mainly include content information, data quality information, citation information, identification information, responsible party information, distribution information, and spatial reference information. Corresponding to the framework model, the metadata entity collection information includes all core metadata of multi-source remote sensing data. The entity MD_metadata is used to represent the metadata entity, and its entity structure is shown in the Figure 5, which specifically includes seven entity sets, namely, MD_ContDesc(MD_ContDesc is used to describe the content information of the EO dataset), DQ_Description(DQ_Description is used to describe the evaluation of the quality of the EO dataset), CI_Citation(CI_Citation is used to describe the reference information of the EO data), MD_Identification(MD_Identification is used to describe the basic information of the EO dataset), CI_RespParty(RespParty is used to describe the responsible party of the EO data), MD_Medium(MD_Medium is used to describe the storage medium of the EO data), and SC_SIRefSys(SC_SIRefSys is used to describe the spatial reference system of the EO data.). Through complete remote sensing metadata, the required remote sensing information resources may be rapidly located.

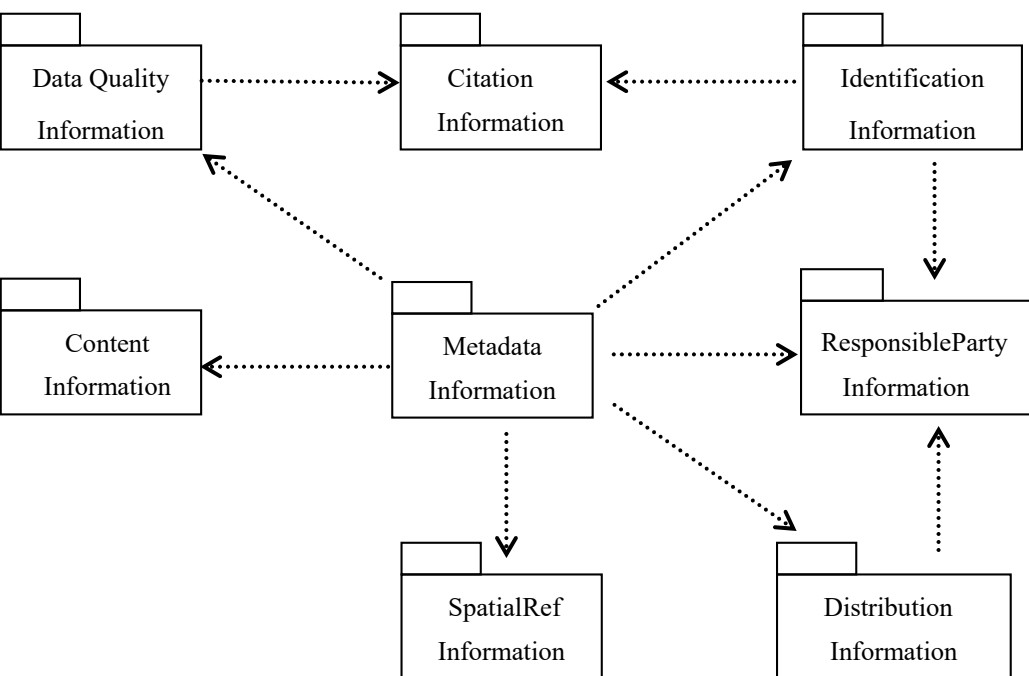

**Figure 4.** Multi-source remote sensing metadata concept structure diagram.

After the above conversion and processing, the metadata and the corresponding remote sensing data are stored in a distributed cluster. When storing data, the remote sensing datasets are divided into several chunks and stored on different shards via the automatic sharding of MongoDB. Because the metadata are document-type data, and the remote sensing data are multi-layer raster data with geographic information, the principle of preservation is different.

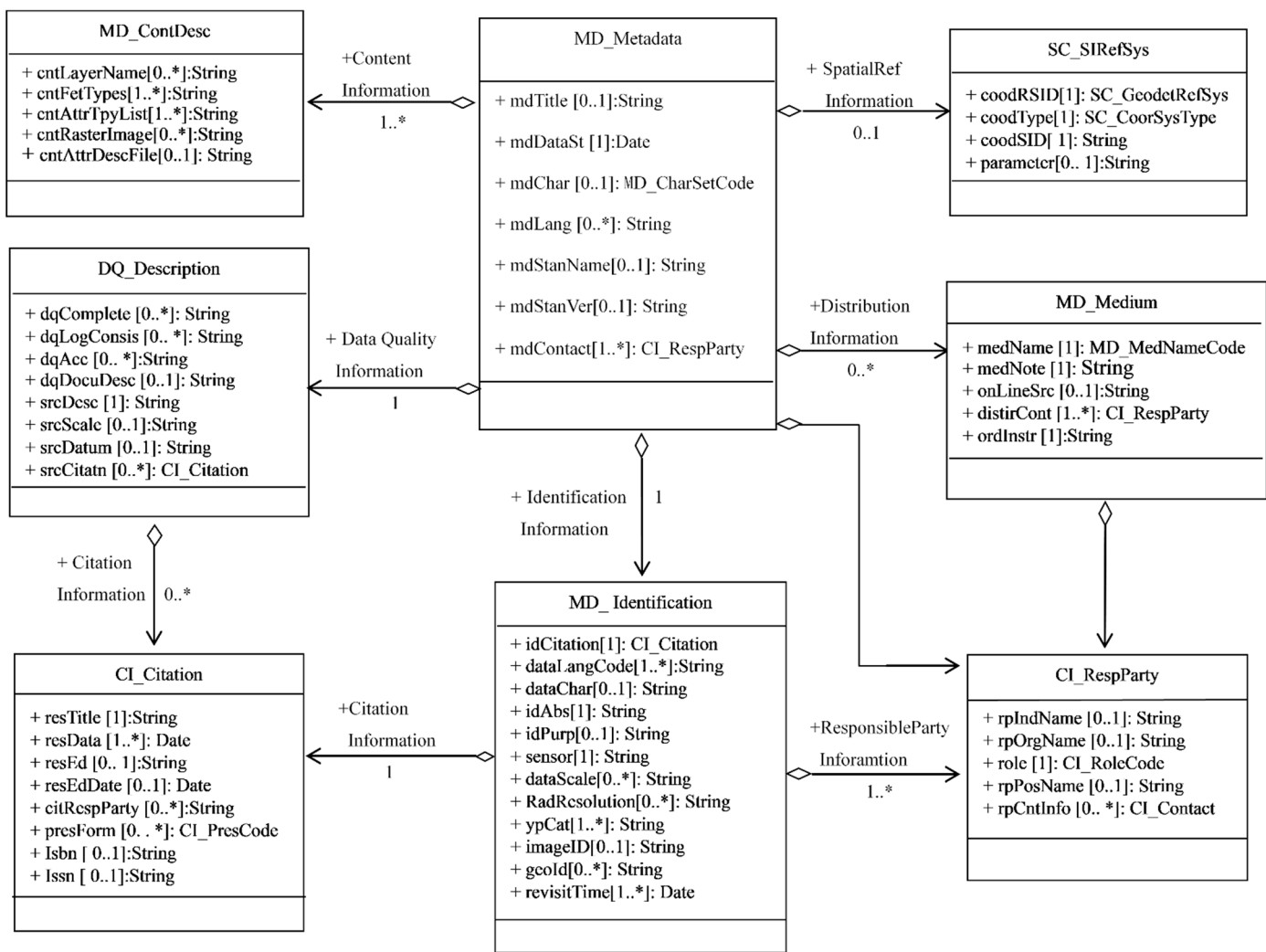

**Figure 5.** Complete multi-source remote sensing information description table.

### 2.4. Distributed Storage of Multi-Source Remote Sensing Data Based on MongoDB

After establishing a spatial management index method for multi-source remote sensing data and designing a complete remote sensing metadata system, massive multi-source remote sensing data should be stored efficiently to achieve the integration of big EO data. Due to the comprehensive improvement of the spatial and temporal resolution of remote sensing data, the traditional spatial data organization model has been unable to meet the existing data requirements. Therefore, the sharding mechanism of the distributed database was used to store multi-source remote sensing data, and its function has mainly been implemented by MongoDB. MongoDB is developed in the C++ language. It is an open source, non-relational database system based on distributed file storage. MongoDB aims to provide scalable high-performance data storage for web applications. However, MongoDB does not have geospatial extension capabilities and cannot be saved directly in the database, unlike document data. This paper develops a unique distributed storage method of remote sensing data based on MongoDB. By setting the relationship between the shard key and the index of chunks, the remote sensing data are stored in different shards in the form of slices, as shown in Figure 6.

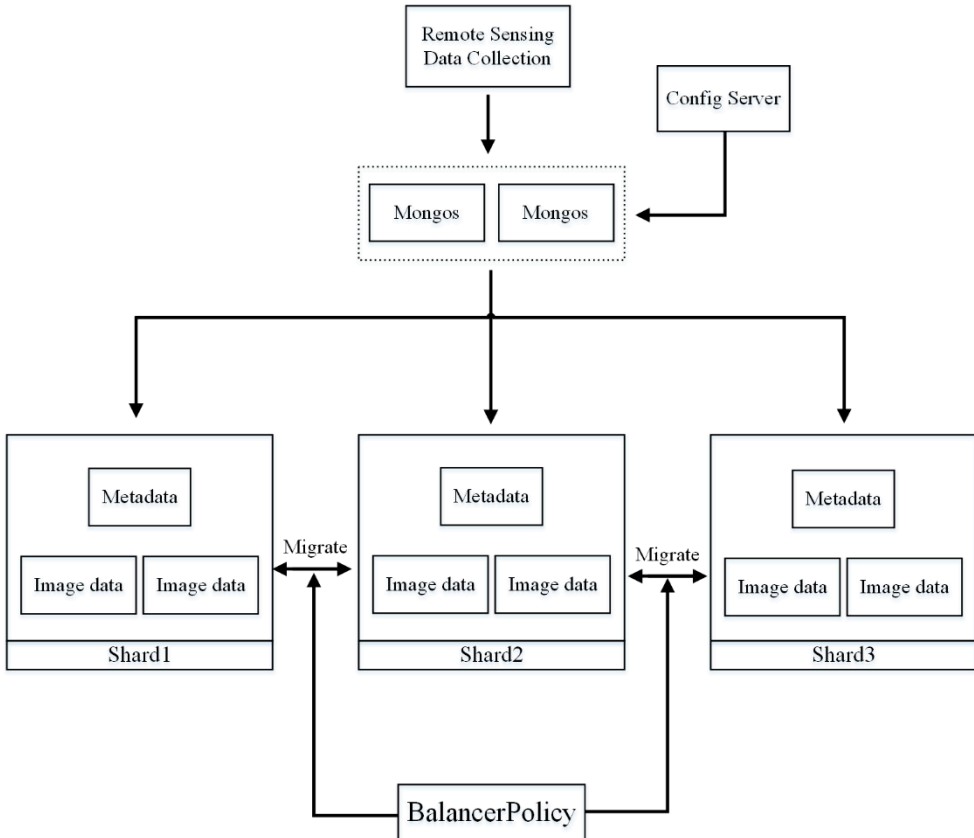

**Figure 6.** Storage workflow in the distributed environment.

The sharding of storage on MongoDB has the following characteristics: (1) after determining the shard key, the MongoDB cluster will automatically partition the data; (2) the balancer process allocates data for each shard and guarantees minimal migration; (3) MongoDB enables flexible setting of chunk sizes. Sharding is a way for MongoDB to horizontally expand the data [22,23]. By selecting the appropriate shard key, the data are evenly stored in the sharding server cluster. The components of sharding mainly include the *mongos* process, the config server, and the shard server cluster. The function of the *mongos* process is to forward the request to the corresponding sharding server. It does not store or process the data. The *mongos* process allows applications to use the MongoDB cluster as a single database instance operation, which facilitates the ease of application development when accessing data.

Each remote sensing dataset finds the corresponding sharding server for reading and writing through the *mongos* process. When an external command initiates a query, the *mongos* process is routed to the specified node to return data. The config server stores cluster metadata, configuration information, and routing information. This includes a collection of instances, such as config.mongos, config.chunks, and config.shards.

The remote sensing dataset is routed to the corresponding shard through the metadata in the config server. The sharding server is the location where data are actually stored in the distributed cluster. Within a sharding server, the remote sensing data will be divided into multiple chunks by MongoDB, and each chunk represents a part of the data inside a sharding server. The remote sensing data and the metadata are divided into multiple chunks, and each record is assigned to a different chunk according to the arrangement of the shard keys. When the size of a chunk exceeds the chunk size set for that configuration, the MongoDB background process will split this chunk into smaller chunks; thus, avoiding the situation where the chunk is too large. In addition, the balancing strategy exists throughout the entire distributed database system to ensure load balancing for each shard. The balancer process will automatically check the distribution of chunks on all shard nodes. When the number of chunks on a shard reaches a certain migration threshold, the balancer process will try to automatically migrate chunks between shards and try to reach the same number of chunks in each shard. This mechanism not only ensures that remote sensing data and the corresponding metadata are efficiently stored in a unified data management system, but also makes full use of unused computer storage resources to achieve the purpose of reasonable storage and rapid retrieval.

## 3. Experiment

The purpose of this experiment conducted here is to test the efficiency of the distributed remote sensing data storage and retrieval based on the SSI model. As stated previously, this framework mainly includes two parts, namely, spatial partitioning and distributed storage. Based on the method of this paper, after the accurate description of the metadata was completed, the remote sensing data were divided into different data slices according to the partition standard of the SSI model. In order to test the effect of the level of efficiency on storing remote sensing data in the database with the SSI model, the same remote sensing dataset has been divided by different levels of grids, and the time consumed by different grid levels of data slice storage in the database has been recorded. Similarly, remote sensing data without SSI model partition have been inserted into a multi-node cluster environment database and standalone mode database, and the performance of the different types of databases has been evaluated. In addition, in order to illustrate the feasibility of efficiently retrieving remote sensing data in distributed environments under the same indexing conditions, a retrieval method based on metadata has been performed in the experiment. Through the data migration method and transmission process in practical applications, the performance evaluation and future plans of this framework have been described.

### 3.1. Datasets and Environment

The purpose of the integration of big EO data is to organize various types of remote sensing data under the same system for management. Various data have different scientific objectives. For example, the Sentinel-2 satellite is a multispectral high-resolution imaging satellite for land monitoring. It can provide images of vegetation, soil, inland, and coastal waters, and can also be used for emergency rescue services. Advanced Spaceborne Thermal Emission and Reflection Radiometer (ASTER) data have a broad prospect in the extraction of mineral alteration information and its spectral range has the characteristics of soil absorption spectrum, which can improve the classification accuracy. At the same time, remote sensing data with the same scientific objective will also have different spatial and spectral resolutions. The experimental remote sensing data of this research work includes that of the Moderate Resolution Imaging Spectroradiometer (MODIS) and ASTER sensors on the TERRA and AQUA satellites, the MultiSpectral Instrument (MSI) sensors on Sentinel-2, and the Operational

Land Imager (OLI) sensors on Landsat 8, which can be downloaded from the following websites: Level 1 and Atmosphere Archive and Distribution System Distributed Active Archive Center (LAADS DAAC) (https://ladsweb.nascom.nasa.gov/search), ESA (https://scihub.copernicus.eu/), and USGS (https://earthexplorer.usgs.gov/). The remote sensing data obtained by each sensor have a complete metadata table to describe the various indicators and characteristics in detail. The data used in this experiment and some of the corresponding metadata are shown in Table 2.

**Table 2.** Source datasets of metadata for experiment. Defense Meteorological Satellite Program (DMSP), Gaofen-1 (GF-1), Huanjing-1A\B (HJ-1A\B), Operational Land Imager (OLI), Enhanced Thematic Mapper (ETM+), Thematic Mapper\ Multispectral Scanner System (TM/MSS), Operational Line-Scan System (OLS), Advanced Spaceborne Thermal Emission and Reflection Radiometer (ASTER), Moderate Resolution Imaging Spectroradiometer (MODIS), Panchromatic and Multi-Spectral_1_2 (PMS_1_2), Charge-Coupled Diode \Hyperspectral Imaging Sensor (CCD/HIS).

| Satellite | Sensor | Spatial Resolution (m) | Spectral Region (µm) | Swath Width (Km) | Revisit Period | Acquisition Year | Data Type |
|---|---|---|---|---|---|---|---|
| Landsat8 | OLI | 15,30 | 0.43–2.29 | 185 | 16 Days | 2013–now | Global Environmental Change |
| Landsat7 | ETM+ | 15,30,60 | 0.45–12.50 | 185 | 16 Days | 1999–now | Natural Resources Research |
| Landsat5 | TM/MSS | 30,120 | 0.45–12.50 | 185 | 16 Days | 1984–2011 | Natural Resources Research |
| Sentinel-2 | MSI | 10,20,60 | 0.4–2.4 | 290 | 10 Days | 2015–now | Global Environmental |
| DMSP | OLS | 500 | 0.4–1.1 10.0–13.4 | 3000 | 101Mins | 1992–2013 | Research on Urban Expansion |
| TERRA | ASTER | 30,60,90 | 0.52–2.43 8.125–11.65 | 60 | 4–16 Days | 1999–2005 | Temperature Retrieval |
| TERRA AQUA | MODIS | 250,500,100 | 0.405–14.385 | 2330 | 1 Days | 1999–now | Integrated Earth Observation |
| GF-1 | PMS_1_2 | 2,8,16 | 0.45–0.89 | 800 | 4 Days | 2013–now | Delicacy Application |
| HJ-1A\B | CCD/HSI | 30,100 | 0.43–0.95 | 700,50 | 4 Days | 2008–now | Environmental Monitoring |

In the experiment, multi-source remote sensing datasets have been partitioned by the SSI model with different levels and then stored in a distributed structure. The metadata in the experiment are inserted into the corresponding data shard, together with the remote sensing image. The remote sensing data integration framework proposed in this paper has been tested by data capture, fragment loading, spatial segmentation, and metadata indexing. By using heterogeneous remote sensing data, the experimental results are mainly related to the level of the spatial grid partition of the remote sensing data based on the SSI model and the degree of metadata indexing. On the basis of the distributed remote sensing data structure proposed in this paper, the time of data management is completely determined by the reasonable degree of spatial index and metadata index of remote sensing data. There are great differences between the organization of multi-node cluster management and standalone mode management for remote sensing data in the distributed environment. Therefore, we have used the remote sensing data in Table 2 in the distributed dataset experiment in the same computer environment for different logical compositions and structures.

The environment of this experiment was completely distributed, and a cluster was created. This cluster was connected by multiple replication sets. Each cluster was composed of three nodes, consisting of a primary node and two secondary nodes. These three nodes were all built with MongoDB version 4.0.9. The primary node contains all operation logs of the remote sensing data, and the secondary node stores all the remote sensing data. The configuration of three the data nodes consists of an i7-4790 (3.6GHz) Central Processing Unit (CPU) and 4.0 GB of Random Access Memory (RAM). Additionally, Python version 3.7 was used as the environment for inserting and querying remote sensing data.

*3.2. SSI Model-Based Remote Sensing Data Storage and Retrieval Experiment*

In this experiment, multi-source remote sensing datasets were stored in a distributed database cluster based on MongoDB through the framework proposed in this paper. All remote sensing data slices based on the SSI model were inserted into a standalone mode database and multi-node database cluster, respectively, and the time consumed to store these data was recorded. The storage experiment process mainly includes two parts, namely, transmitting remote sensing data slices and allocating storage resources. After the storage experiment, the retrieval experiment was performed on the remote sensing data that had been inserted into the distributed database. The OLC code was used in the metadata table as an index to measure the time consumed in different environment.

Figure 7 shows the storage of remote sensing data in a MongoDB cluster. The input is the remote sensing data slices that have been spatially partitioned based on the SSI model. These data slices have been assigned spatial index codes and are linked to corresponding metadata tables. After the data slices were inserted into the computer memory space, the *mongos* process allocated different slices to the corresponding shards via the shard's information and the slice information in the config server. A reasonable shard key is used as an index by the MongoDB cluster, and the input data were allocated by the principle of the shard key. After receiving the request, the *mongos* process transferred the data to the shard. There are two possibilities for each shard: (1) as the volume of data increases, the size of the datum chunks in shard A will exceed the configured chunk size. In general, the default value is 64 M. When the chunk exceeds the threshold, the chunk is split into two identical chunks. The growth of data will make the chunks split more and more. (2) When the number of chunks on each shard is unbalanced, the balancer component in the *mongos* process will execute the automatic balancing strategy. Here, the process moves the chunk from the shard B with a larger amount of data to the shard C, which has the least amount. Chunks in MongoDB will only split and will not merge. Therefore, if the chunk size is modified to be larger, the number of existing chunks will not be reduced, but the chunk size will continue to increase with data inserted until the target size is reached. Through this storage method, the computer resources of each data shard are not wasted, thereby achieving the purpose of load balancing. In addition, when data are transmitted to the distributed cluster, while the shard in it is down or offline, the data can be stored in other shards first. After the fault is eliminated, the *mongos* process will retransmit the data back to the original shard. In this way, it can not only provide high efficiency for data storage, but also ensure the security of data during transmission. Multiple shards can form a complete MongoDB cluster.

In order to test the storage efficiency of remote sensing data in different environments and the impact of different levels of shards on the storage speed, based on the SSI model, four sets of remote sensing data with successively increasing capacities were selected as experimental data. Four groups of data were inserted into a multi-node database cluster and a standalone database to verify the performance of the proposed method. The four groups of data were partitioned by the SSI model into "Level 2", "Level 3", and "Level 4", and the experiments were divided into 12 groups and three large groups of experiments, using such experiments to test the impact of the partition level on the performance of the framework. In addition, subsequent query experiments have been performed on the basis of this experiment.

In the storage experiment, remote sensing data were spatially partitioned according to different levels and stored in a standalone database and multi-node database cluster, respectively. The retrieval experiment was implemented in two different environments through the OLC identification code in the metadata. In a multi-node cluster, when an external request initiates a data retrieval, in order to get an appropriate retrieval response, the mongos process has to interact with multiple shards. Finally, the *mongos* process is automatically routed to the specified node based on the data distribution and returns the retrieval results. The purpose of this experiment is to test the retrieval performance of the structure with shards, which will be more advantageous than the retrieval standalone mode.

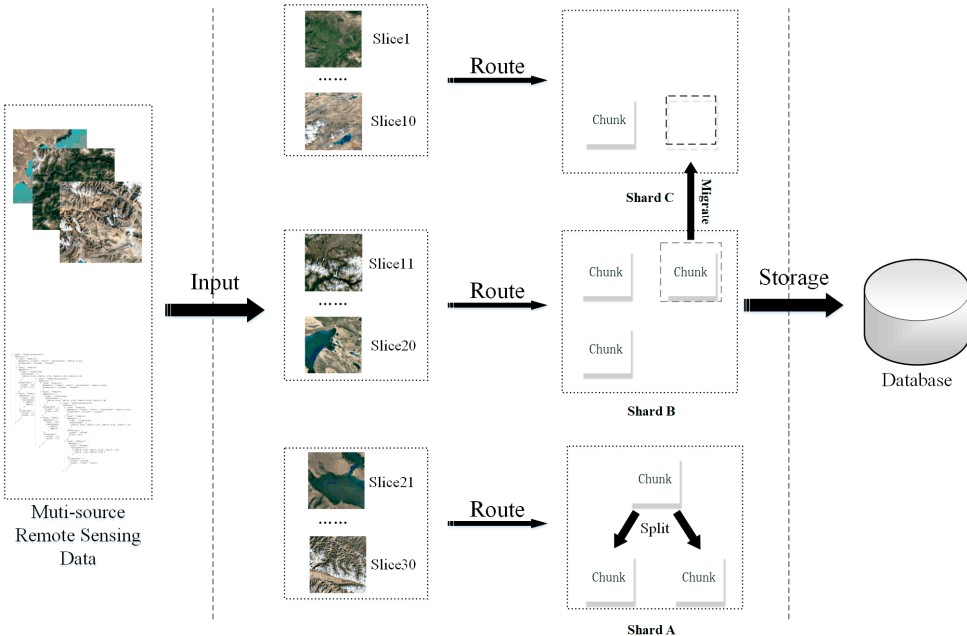

**Figure 7.** Remote sensing data distributed storage process.

## 4. Experiment Result and Analysis

The efficiency of storing remote sensing data in a multi-node cluster is indeed higher than that of standalone mode, as shown in Figure 8. With the increase of remote sensing data, the time consumed continues to increase. When the amount of remote sensing data is greater than 40 GB, the growth rate of time consumption in the standalone mode remains the same, but the growth rate in distributed mode slows down gradually. The purpose is to distribute multiple business logics of a single storage request to multiple nodes, so that multiple logical commands can be processed simultaneously. This result obviously shows that the proposed remote sensing data integration framework is feasible. In this experimental group, as the amount of data increases, the time consumed by storing data also increases, as shown in Figure 9. At the same time, as the grid level of the SSI model increases and the time consumed also increases. However, when the grid level is higher, the time increase is not obvious. For the same data, the storage time of a grid based on Level 4 is longer than that based on Level 2. The reason is that the rise of the grid level leads to more loose data, so the *mongos* process spends more computer resources and time to allocate data. This distributed storage mode is a parallel processing flow, which is related to the running nodes. This experiment is a three-node MongoDB cluster. The data partition of the "Level 4" grid reaches the threshold of the cluster processing dataset in this experiment, so the time consumed is slightly longer than the other levels. However, the cluster established in this experiment can process Level 2 and Level 3 data blocks. Therefore, the time for storing Level 2 and Level 3 data reaches a relatively stable state.

When the amount of remote sensing data is relatively small, the number of nodes in the cluster and the grid level of the SSI model have little effect on the efficiency of retrieval. As the amount of remote sensing data gradually increases, the time consumed by the retrieval work also gradually increases. The reason for this is that the increased data causes more records to be traversed by the *mongos* process. Under the same conditions of the amount of remote sensing data, as grid level in the SSI model increases, the time consumed by the retrieval is gradually reduced. At the same time, the retrieval experiment and storage experiment have the same trend, i.e., when the grid level in the SSI model is increased, the time consumed by the retrieval will reach a relatively balanced state, which is consistent with the results of different grid level storage experiments.

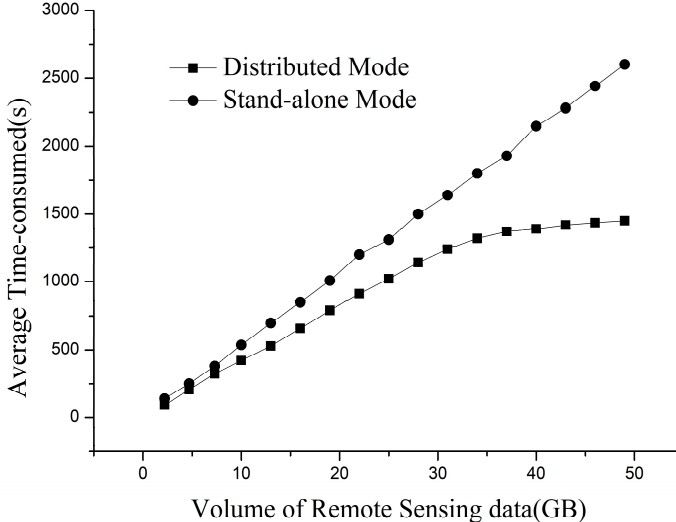

**Figure 8.** The result of storage experiment.

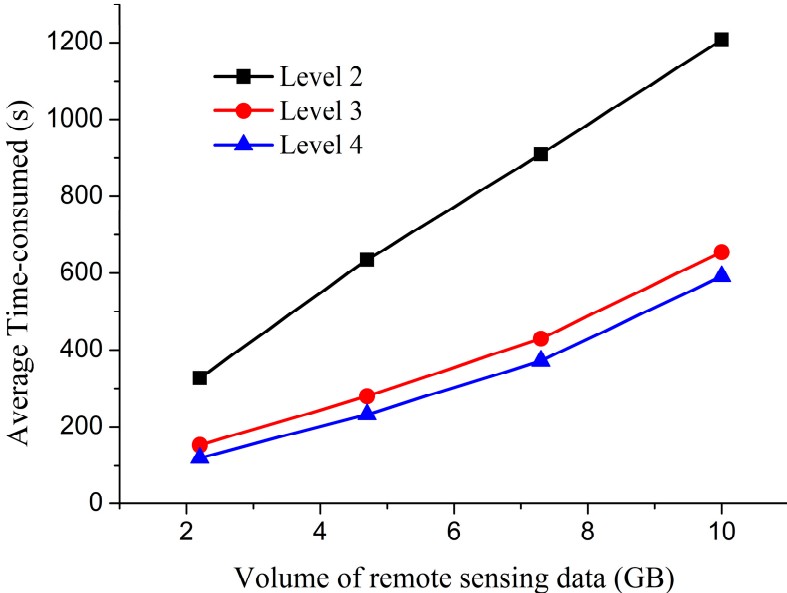

**Figure 9.** The result of storage experiment with different Level on SSI model.

As can be seen in Table 3, in the same situation, the retrieval speed of the multi-node cluster is slightly faster than that of standalone database, and the advantages are not obvious. The main reason is the choice of shard key which is also crucial for retrieval speed. The quality of the shard key determines whether the data are evenly distributed in shards. If a proper slice key is selected, data will be written to only one shard. Data will not be inserted into other shards until the shard threshold is reached. Retrieval in a multi-shard cluster will cause the *mongos* process to merge and sort the results. Because data sharding is oriented to distributed storage, the retrieval process is parallelized. Considering the use of parallel processing, "MapReduce" functional modules will be evaluated in future research work.

**Table 3.** The result of retrieval experiment.

| Data Volume (rows) | SSI Model Level | Query Time on Distributed Environment | Query Time on Stand-Alone Mode Environment |
|---|---|---|---|
| 9100 | Level-2 | 0.02 s | 0.02 s |
| | Level-3 | 0.02 s | 0.02 s |
| | Level-4 | 0.02 s | 0.02 s |
| 20000 | Level-2 | 2.31 s | 2.43 s |
| | Level-3 | 1.19 s | 1.17 s |
| | Level-4 | 0.89 s | 0.95 s |
| 30000 | Level-2 | 6.02 s | 6.16 s |
| | Level-3 | 2.53 s | 2.68 s |
| | Level-4 | 2.40 s | 2.53 s |
| 45000 | Level-2 | 11.64 s | 11.73 s |
| | Level-3 | 5.07 s | 5.03 s |
| | Level-4 | 4.66 s | 4.82 s |

## 5. Conclusions and Future Work

In order to solve the problem of the integrated management of big EO data, this paper proposes a distributed multi-source remote sensing data management framework based on MongoDB and the SSI model. In the framework, in order to express the geospatial area range and spatial position covered by multi-source remote sensing data uniformly and establish an efficient remote sensing data organization and spatial indexing method, the SSI model has been proposed as a spatial index of remote sensing data. Meanwhile, we have innovatively used the GeoJSON data structure as the storage form of remote sensing metadata and designed a complete multi-source remote sensing metadata system to provide an index foundation for efficient data retrieval. The core part of the proposed framework is a distributed storage cluster of remote sensing data, which provides a secure and stable data storage method for the integration of remote sensing data. At the same time, a distributed data structure of remote sensing data based on MongoDB was used as a data storage method in the cluster. The combination of the data structure and spatial grid partition based on the SSI model can implement the retrieval, integration, and sharing of big EO data. Additionally, in the case of high concurrency, multiple requested queries may be combined into one run to reduce the number of database queries. The purpose of this framework is not only to provide a distributed management idea for the integration of multi-source remote sensing data, but also to provide a theoretical data structure foundation for the parallel computing of big EO data.

Experiments have been designed to verify the feasibility of the framework and evaluate the performance of remote sensing data integration and retrieval. This paper has made a comparison of the time required to insert the same volume of remote sensing data into a standalone database and a multi-node database cluster. The results showed that the integrated management method of remote sensing data with the distributed mode had better performance. At the same time, with the increase of the level of spatial grid partition on the SSI model, the time consumed by data storage will increase and reach a stable state. However, the level of the grid cannot be reduced in order to reduce the storage time, because the spatial range of remote sensing data required is different. We also used the OLC code in the SSI model as the retrieval foundation to test the performance of standalone database and multi-node database cluster. As a result, the speed in the distributed environment was slightly faster than the storage method in the standalone environment. Similarly, with the increase of the level of spatial grid partition in the SSI model, the time consumed in querying data will be reduced and reach a stable state, which is consistent with the storage experiment. The reason for this may be due to the existence of an automatic balancing mechanism, where remote sensing data are accelerated to be allocated to different shards, and the sharding mechanism does not work during retrieval; hence, the

retrieval speed in a distributed environment is not changed. In addition, the factors of insufficient data and the selection of the shard key are also important reasons.

This article has basically implemented the integration of multi-source remote sensing data but has not yet applied the theory of big EO data to practical research work. Future work will be based on the parallel computing of big EO data and spatial data mining. In addition, authors of this paper have also performed parallel preprocessing on vector data in a shapefile format. Since integrating many different data in the same environment requires many pre-processing steps, it can take a lot of time and computer resource costs. Therefore, future work will focus on the spatial data with distributed data structures, combined with high-performance algorithms and parallel computing environments in order to provide a scientific basis for the use of big data in different fields.

**Author Contributions:** K.Z. and J.W. conceived and designed the research; Y.C. and J.Y. implemented the method and performed the experiments; all the authors reviewed and edited the manuscript. All authors have read and agreed to the published version of the manuscript.

**Funding:** This study was funded by the National Key R&D Program of China (2018YFC0604001-3), B&R Team of Chinese Academy of Sciences (2017-XBZG-BR-002), and National Natural Science Foundation of China (No.U1803117, No.U1803241).

**Acknowledgments:** We would like to thank Xinjiang Laboratory of Mineral Resources and Digital Geology of the Chinese Academy of Sciences for guidance and full support.

**Conflicts of Interest:** The authors declare no conflict of interest.

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
