# Peer review of "Big Earth Observation Data Integration in Remote Sensing Based on a Distributed Spatial Framework"

_remotesensing, doi:10.3390/rs12060972_

Round 1

Reviewer 1 Report

General comment

Extensive punctuation check required.

Specific comments

Title:

Extra space before “Distributed”

Introduction:

Last paragraph, first line: change “distribution” to “distributed”

Section 2.2

3rd paragraph, 4th line: Remove hyphen after “grid [23]”.

Figure 4 missing.

Correct these for Figure 4:
Typo in “Ciation Information” between DQ_Description, CI_Citation and MD_Identification.

Explain in the caption what are the diamond-shaped arrowheads in the figure. 

Section 2.4

First paragraph, 6th line: remove double commas after “Therefore”.

Section 3.1

First paragraph, 5th line: The line starting with “While..” is incomplete. Correct this.

First paragraph, 9th line: change “experiment” to “experimental”

Missing period before “(MODIS)”
extra space after “DAAC”

Figure 7 Caption:

Add period at the end and correct the fonts.

Experiment Result and Analysis

Remove character after “4.” on the heading.

Correct the line spacing for the second paragraph.

Conclusions and Future works

Last paragraph
4th line: change “also have” to “have also”

4th line: remove “However”

5th line: remove “condition”

6th line: remove extra space after “costs.”

References

Missing period for reference entry 3. 

Reviewer 2 Report

Only comment is that table 5 was missing. Thank you for addressing my concerns about the English in the manuscript.

Reviewer 3 Report

An interesting paper in a specialised sector. The paper is clear in explaining the SSI and then applying it to selected data, with time results being shown.

In figure 9 the x-axis should have "volume" not "volumn".

Author Response

This manuscript is a resubmission of an earlier submission. The following is a list of the peer review reports and author responses from that submission.

Round 1

Reviewer 2 Report

The manuscript has identified a need to speed up access to multi source remote sensing data, at different temporal and spatial scales.  While the manuscript indicates that their approach does speeds accessing, what is not clear from this work is the quality of the recovered data?  This needs to be addressed before recommending publishing as just because it is quick does not guarantee the quality of the final product. 

There are quite a few grammar mistakes that need to be corrected, but also please define acronyms before you use them.  SSI is used in the abstract but is not defined till the third page

Earth should always be written as Earth, earth is referring to soil.

On Line 55 you mention will reach petabyte level but do not indicate when which is missing from the sentence making sense.

Line 83: you need either a period or a semi-colon before However, if you need a comma here then however should be lower case

Figure 3 is in the wrong place and make reading this section quite difficult.

Reviewer 3 Report

The authors propose a new framework to solve data structure problems in implementing multi-source remote sensing data integration.
My concerns are:
- the text is difficult to read because is not objective. Please go through the manuscript carefully to correct grammar misuses, typo errors, and style (repetitions). Figure 3 is in the wrong order. Up to page 6 lines are numbered but not in the rest of the manuscript;
- the term "framework" is not precisely used. It should refer to an abstraction around which a specific implementation could be developed. In this work, there is no discussion about the extensibility points of the framework. It does present a specific implementation based on Open Location Code and MongoDB;
- considering this, the review of related work is incomplete since there are published articles describing remote sensing data and MongoDB;
- at the core of the proposed framework is the spatial segmentation indexing model, which is simply the use of the Open Location Code method. The method was created by a third party, but still, the authors make a long description of the method in section 2.1. It is not better than the reference to the method original reference.
- misunderstanding of the concept of spatial indexing. It is not clear why the authors argue that using OLC is an improvement in spatial indexing;
- about the Metadata standard, the authors use a subset of the ISO 19115-2, and again provide a long description of it in section 2.2, adding nothing particular to this work. Using GeoJSON to codify the metadata is something very common.
- the authors use the terms "multi-node cluster" and "stand-alone mode". Shouldn't it be single-node?
- the experiment is based on data insertion and retrieval. However, it is not clear how the retrieval strategy is not described (e. g. randomly, spatially aligned, by different regions of interest). The strategy affects the results and should be considered to improve the experiment.
- the extensive use of the term integration. what the authors consider the integration of multi source remote sensing data is not clear. It seems that is just having the data organized in the same structure.
Finally, the authors have not discussed the cons of their approach. For example the costs of having all the data preprocessed to the same spatial reference system.